# Designing Safe General LED Lighting that Provides the UVB Benefits of Sunlight

**Seung-Taek Oh [1], Dae-Hwan Park [2] and Jae-Hyun Lim [2,*]**

[1]  Smart Natural Space Research Center, Kongju National University, Chungcheongnam-do 31080, Korea; ost73@kongju.ac.kr
[2]  Dept. of Computer Science & Engineering, Kongju National Uiniversity, Cheonan-si, Chungcheongnam-do 31080, Korea; glow153@gmail.com
*  Correspondence: defacto@kongju.ac.kr; Tel.: +82-41-521-9558

**Abstract:** The ultraviolet (UV) rays emitted from sunlight greatly influence human health. Excessive exposure to UV rays can be harmful to eyes and skin; however, limited UVB exposure is essential for the synthesis of vitamin D. Nowadays, owing to insufficient exposure to natural light, there is increasing concerns about low vitamin D amongst individuals. To address this issue, many lighting devices that provide UVB doses have been released; however, such devices are only used for treatments or for special purposes. This study proposes a general indoor lighting system with a UVB LED light source to provide safe UVB doses to users who spend large amounts of time indoors. The optical characteristics of two UVB LEDs with output of 20 and 100 mW were analyzed based on their distances and applied currents. The light source combination of UVB LEDS that meets the UV hazard standard of IEC-62471 was derived; this is a photobiological safety evaluation standard of LED lighting devices. We then produced a lighting module in which the UVB LED light source was applied to general LED lighting and measured and analyzed the spectral irradiance of the proposed lighting according to the measurement standard for the general lighting of IEC 62471. The actinic UV hazard (AUV) and near-UV hazard (NUV) were calculated to be 0.001 and 10 W/m$^2$, respectively. Thus, the provision of UVB dose did not pose any risks. In addition, the total EUV (Erythemal weighted UV) dose when the proposed lighting was implemented for 16 h was 187.66 J/m$^2$, confirming that this dose did not cause erythema for the general skin types (Skin Types 1–6). Further, the design plan of general indoor lighting with a UVB LED light source is presented.

**Keywords:** UVB; UVB LED general lighting; UV safety evaluation; general lighting design; UVB characteristics of sunlight

## 1. Introduction

Ultraviolet (UV) rays are electromagnetic waves of 200–400 nm wavelength bands comprising UVA (315–400 nm), UVB (280–315 nm), and UVC (200–280 nm) [1]. Of these, the UVC and most UVB rays are absorbed into the ozone layer, and 10–30% of UVB and UVA, respectively, reach the earth's surface and affect humans [2]. Excessively exposure to UV radiation causes a harmful effect on the human body. UVA promotes aging on the skin, while UVB can cause skin cancer [3,4]. Any wavelength lower than 315 nm has negative effects on the eyes, causing inflammation of the cornea or cataract [5]. However, adequate exposure to UVB is beneficial for human health as it assists in the synthesis of vitamin D and promotes mental health [6]. UV of sunlight can be harmful or beneficial depending on the degree of exposure [7], and issues on deterioration of health, such as vitamin D deficiency due to a lack of exposure to natural light, have been raised recently [8,9]. Vitamin D deficiency causes rickets, osteomalacia, and hypotonia and increases the risk of developing diabetes, hypertension,

and myocardial infarction [10,11]. Therefore, vitamin D is essential for human health. Studies have introduced several ways to supplement vitamin D [12–14]. Vitamin D in the body is produced through ingestion of food and supplements, or skin exposure to UVB. However, vitamin D supplementation through diet has been reported to be limited [12,13,15]. Even though one steadily ingests foods beneficial to produce vitamin D, such as milk and salmon, or vitamin D supplements, vitamin D deficiency is still found [16]. Meanwhile, since sunlight can support vitamin D production even with a short exposure to it [15,17], exposure to the UVB of the sunlight is suggested as the most effective method of meeting the vitamin D requirements [14]. However, many people stay indoors for an average of 85–90% a day [18], and some of the countries in the northern hemisphere, such as the USA and European countries, are raising the issue of a shortage of sunlight due to geographic factors. In many cases, the decrease of outdoor activities is caused by an averting behavior caused by the deterioration of air quality due to fine dust or smog. In the United States, when the air quality index (AQI) is high, the outdoor activity time is reduced by 18%—among the elderly, by 59% [19,20]. In Korea, an alert service is operated to encourage people to refrain from outdoor activities when the concentration of fine dust is high [21]. Therefore, natural light cannot completely satisfy the required UVB dose. To address this issue, UVB irradiation devices that artificially provide adequate UVB doses have been proposed [22]. However, such UVB irradiation devices are limited by the fact that there are special lightings to which certain parts of the body must be exposed at certain distances and are used for treatments with limited uses [23]. Specialized lighting for treatment does not provide natural UVB light in daily life like sunlight because users must consciously use instruments after spending money and time. Therefore, there is a lack of research on the development of general lighting that provides UVB for human health for people who are not frequently exposed to natural light.

Recently, LED light sources capable of providing various optical characteristics of UV and IR and the visible light characteristics of natural light have been developed. Although LEDs are a low-power, high-efficiency light source with a high manufacturing cost, various optical characteristics can be reproduced via output control for industrial applications [24,25]. Several lighting devices employ LED, such as light-therapy facemasks, those found in plant factory and micro algae cultivation facilities, and lighting devices for sterilization and environmental purification using UV LEDS [26]. Owing to these devices, the concern about the harmfulness of lighting devices to the human body has increased. Therefore, the International Electrotechnical Commission (IEC) introduced IEC 62471, a standard for the photobiological safety evaluation of lighting: in Europe, all lighting devices must display their own IEC-based hazard rating [27,28]. The hazard level of UV-based devices was evaluated based on the IEC 62471 standard for safe use [29].

This study proposes a design plan for indoor lighting that can provide the UVB dose safely to indoor users using a UVB LED light source that can control the current output. The optical characteristics of two types of UVB LED light sources of 20 and 100 mW output specifications were measured and analyzed, respectively, based on their distances from the lighting source and applied current. In the analysis process, the actinic UV hazard radiation (AUV, 200~400 nm) and near-UV hazard radiation (NUV, 315~400 nm) values that are used to determine the UV hazard level based on IEC 62471 were calculated; then, a combination of light sources that enables safe UVB radiation was derived. Finally, in order to support the synthesis of vitamin D, we produced a general lighting module in which UVB LEDs added to general (visible) LEDs was manufactured and measured, and the spectral irradiance in the lighting box shielded from external light was analyzed. After confirming whether the proposed lighting causes erythema on human skin, it was verified whether the UV hazard standard of IEC 62471 was followed. The UVB characteristics of sunlight were reproduced via this method, and a development plan for safe, general indoor lighting that enables vitamin D synthesis is suggested.

## 2. Methods: UVB Characteristics of Natural Light

UVA and UVB are components of UV rays that have different levels of effects on the human body [30]. The local solar UV irradiance was measured and analyzed to provide a typical solar irradiance used to calculate UV exposure. Measurements were taken at a latitude of 36.85 and a longitude of 127.15 on the roof of a 10-story building to minimize shadow interference. The spectral irradiance was measured at noon on a clear day in summer since it would be advantageous to identify the characteristics of UV light under strong sunlight. The measurement was conducted on 21 July 2018, and the concentrations of the fine dust were 76 µg/m$^3$ for pm10 and 50 µg/m$^3$ for pm25, which are "normal" levels (according to the Korea Meteorological Administration). A spectrometer (CAS 140CT 152) capable of measuring spectral irradiance in the range of 200–800 nm was used as the measuring instrument. Its specifications are shown in Table 1. In addition, the results of the measured natural light are shown in Figure 1.

**Table 1.** Specifications of the spectrometer (CAS 140CT 152).

| Model (Company) | Image | Spectral Range [nm] | Number of Pixels | Resolution [nm] |
|---|---|---|---|---|
| CAS140CT 152 (Instrument Systems) | | 200–800 | 1024 × 768 | 2.7 |

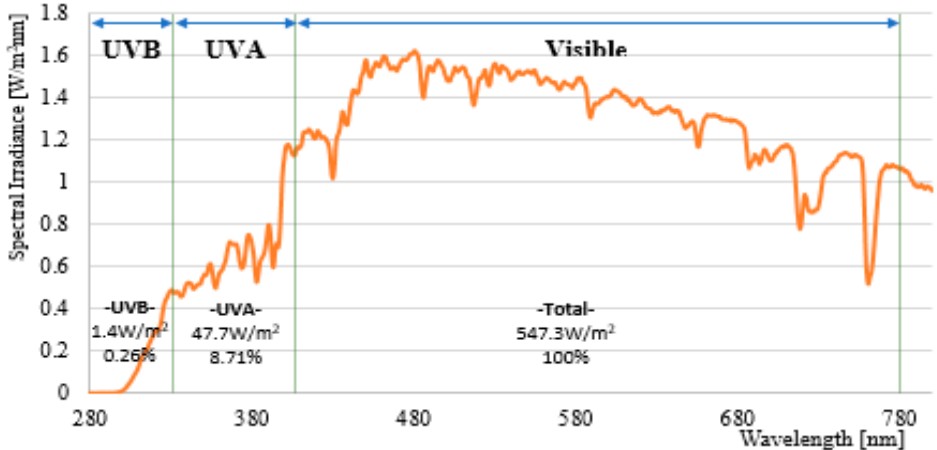

**Figure 1.** Characteristics of sunlight by wavelength.

The sunlight characteristics based on the measured results are shown in Figure 1. We calculated the total irradiance of sunlight, UVA, and UVB irradiance using Equation (1). Erythemal weighted UV radiation (EUV), which is UV radiation that produces erythema at a given wavelength, was calculated using Equation (2). EUV is a weighted measure of the likelihood of erythema, a type of skin burn occurring in the UV wavelength band, which is needed to calculate safe levels of exposure to ultraviolet light and to calculate the amount of vitamin D synthesis [14]. At this time, the erythema weighting function ($S_{er}$) of Equation (3) was applied according to the definition of Commission Internationale de l'Eclairage (CIE) [31,32].

$$\mathrm{UVA} = \int_{315}^{400} E\lambda d\lambda \ \left(\mathrm{W/m^2}\right), \ \mathrm{UVB} = \int_{280}^{315} E\lambda d\lambda \ \left(\mathrm{W/m^2}\right) \tag{1}$$

$$\mathrm{EUV} = \int_{280}^{400} E\lambda S_{er} d\lambda \ \left(\mathrm{W/m^2}\right), \ \mathrm{EUVA} = \int_{315}^{400} E\lambda S_{er} d\lambda \ \left(\mathrm{W/m^2}\right), \ \mathrm{EUVB} = \int_{280}^{315} E\lambda S_{er} d\lambda \ \left(\mathrm{W/m^2}\right) \tag{2}$$

$$S_{er}(\lambda) = \begin{cases} 1 & (250 \leq \lambda \leq 298) \\ 10^{0.094(298-\lambda)} & (298 \leq \lambda \leq 328) \\ 10^{0.015(140-\lambda)} & (328 \leq \lambda \leq 400) \end{cases} , \ (\lambda : \text{Wavelength[nm]}). \quad (3)$$

Thus, the total irradiance was 547.3 W/m$^2$, of which UVA and UVB were 47.7 and 1.4 W/m$^2$, respectively, accounting for 8.71% and 0.26% of the total irradiance; thus, the spectral irradiance of UVA was 35 times higher than UVB. However, the results of applying the erythema weighting function were 0.05 and 0.15 W/m$^2$ for EUVA and EUVB, respectively. Thus, EUVB was three times higher than EUVA, and UVB had a considerable effect on the human body compared with UVA. EUV was 0.2 W/m$^2$, which can cause erythema upon exposure for longer than 25 min if the person has a skin color of beige to light brown [33]. A previous study revealed that Asians (Koreans) could produce 200 IU of vitamin D when exposed to an EUVB intensity of 0.15 W/m$^2$ for ~12 min (720 s) [34]. Therefore, in order to support more than 200 IU of vitamin D synthesis through lighting, EUVB doses greater than 0.15 W/m$^2$ × 720 s = 108 J/m$^2$ (Joule = Watt × Time [35]) could be provided to skin types commonly found in the Asian region.

## 3. Methods: UVB LED General Lighting Design

### 3.1. UV Safety Evaluation

Excessive exposure to the UV rays can cause erythema, which is a kind of skin burn [36], and UV can be harmful to the human body even if a person is exposed to the UV at a lower level continuously [7]. Therefore, the UVB-LED general lighting proposed in this paper provides a UVB intensity of light at a level that would not cause erythema, meeting the safety evaluation standard (IEC 62471) for lighting devices.

The IEC has established the IEC 62471 standard to evaluate the photobiological safety of lighting devices using LED light sources [37]. Therein, the hazards caused to eyes and skin due to lighting are divided into the following eight categories: eye and skin photochemistry UV, eye UVA, retinal blue light, retinal blue light with a small light source, retinal heat, retinal heat caused by weak visual stimulus, eye IR, and skin heat. Each item is classified and marked by four categories, namely, the exclusion group, the low-risk group, the moderate-risk group, and the high-risk group, after analyzing spectral irradiance and radiance measured by wavelength range. IEC 62471 also classifies lighting into a general lamp service (GLS) and other lamps, and applies different test methods accordingly. In this case, GLS refers to general lighting used at homes or offices, and other lamps, excluding GLS, are those used for industrial or medical lighting devices used for suntan, water purification facilities, and medical treatments [23]. Depending on the type of lamps used, the optical characteristics of the GLS are measured with a 500-lux brightness. We further evaluated whether a hazard caused by the UVB dose existed. At this time, the hazard assessment based on the UVB light source is limited to an actinic UV hazard (AUV) and a near-UV hazard (NUV). Details of emission limits for each item are summarized in Table 2 [37].

**Table 2.** IEC 62471 classification standard for ultraviolet (UV) risk items.

| Hazard Items | Unit | Exempt | Risk Group 1 | Risk Group 2 |
|---|---|---|---|---|
| Actinic ultraviolet hazard (AUV) | W/m$^2$ | **0.001** | 0.003 | 0.03 |
| Near-UV hazard (NUV) | W/m$^2$ | **10** | 33 | 100 |

Exempt is the safest lamp rating in Table 1, and its radiation in each item of AUV is 0.001 W/m$^2$. NUV is applied when Exempt $\leq$ 10 W/m$^2$. AUV is calculated by integrating the AUV hazard

weighting function ($S_{UV}$) to the spectral irradiance of 200–400 nm wavelength band (Equation (4));
NUV was calculated by integrating the spectral irradiance of 315–400 nm UVA wavelength band.

$$\text{AUV} = \int_{200}^{400} E(\lambda)S_{UV}(\lambda)\mathrm{d}\lambda \tag{4}$$

where $\lambda$ = wavelength, $E(\lambda)$ = UV spectral irradiance, and $S_{UV}(\lambda)$ = actinic ultraviolet hazard
weighting function.

In this case, the AUV hazard weighting function ($S_{UV}$) uses the weight table value by wavelength
according to IEC 62471 [28]. When no reference value for the observed spectral wavelength is available,
a weighted value based on each wavelength is calculated by linear interpolation, as shown by $S'_{UV}(\lambda)$
in Equation (5).

$$S'_{UV}(\lambda) = S_{UV}(\lambda_l) + (\lambda - \lambda_l)\frac{S_{UV}(\lambda_r) - S_{UV}(\lambda_l)}{\lambda_r - \lambda_l}, \ (\lambda_l < \lambda < \lambda_r). \tag{5}$$

Equations (6) and (7) are the final formulas of AUV and NUV, respectively.

$$\text{AUV} = \sum_{\lambda=200}^{400} E(\lambda)S'_{UV}(\lambda)\Delta\lambda \tag{6}$$

$$\text{NUV} = \int_{315}^{400} E(\lambda)d\lambda = \int_{315}^{400} E(\lambda)\Delta\lambda. \tag{7}$$

### 3.2. Overall Lighting Design Using the UVB LED Light Source

IEC 62471 states that the stability evaluation of general lighting must be measured at an
illuminance of 500 lux [27]. Further, considering that the average ceiling height of houses, apartments,
and offices are 2.4–2.8 m, general lighting should be able to provide a brightness of 500 lux at a distance
of 1.5 m. To this end, white multi-channel LEDs are used for such purposes. UVB LED light sources
were utilized to provide UVB doses that are beneficial to the human body. LEDs can control the optical
characteristics by adjusting the applied current output. LED light sources may be combined to form
various types of lighting. Based on these characteristics, to control all LED light sources, a controller
was developed so that the desired brightness and the UVB dose could be achieved. Figure 2 outlines
the proposed lighting.

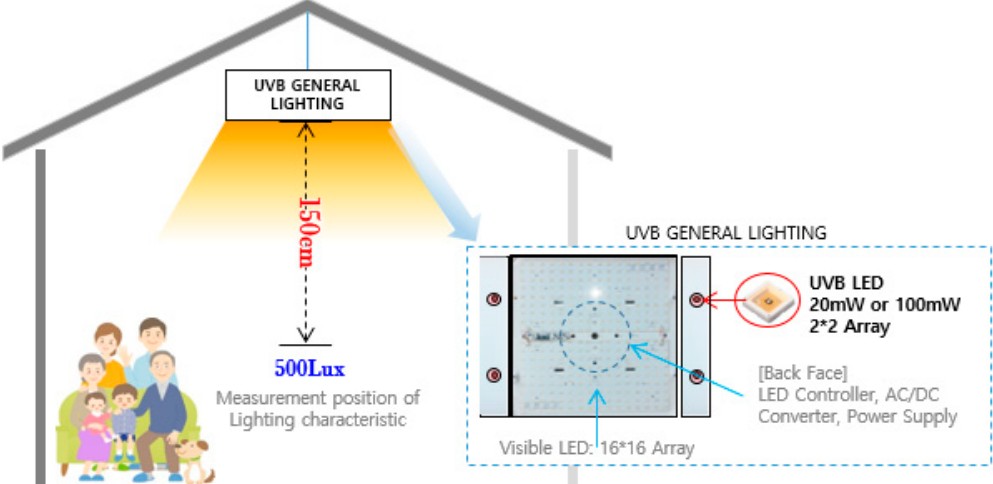

**Figure 2.** Outline of indoor lighting with UVB LEDs.

Two types of UVB LED light sources (LG InnoTek, Seoul, Korea) released in Korea were selected to provide safe UVB doses; their optical characteristics based on distances and applied currents were measured and analyzed. The two types of UVB LED light sources used in the experiment are shown in Table 3. The UVB LED light sources had an output power of 20 and 100 mW, respectively, and a current up to 350 mA could be applied.

**Table 3.** Specifications of UVB LEDs (LG InnoTek).

| Model Name | Size (L*W*H) (mm) | Peak Wavelength | Optical Power (mW) | Maximum Current [mA] |
|---|---|---|---|---|
| LEUVA66G00KV00 | $6.0 \times 6.0 \times 1.35$ | 295~315 nm | 20 | 350 |
| LEUVA66H00KU00 | $6.0 \times 6.0 \times 1.35$ | 300~310 nm | 100 | 350 |

For the experiment, two test lighting modules combining 10 light source modules (each 20 mW UVB led or 100 mW UVB led) were developed. The light characteristics of the two UVB LED light source modules were measured from distances of 30, 60, 50, 90, and 150 cm, and the results are shown in Figure 3.

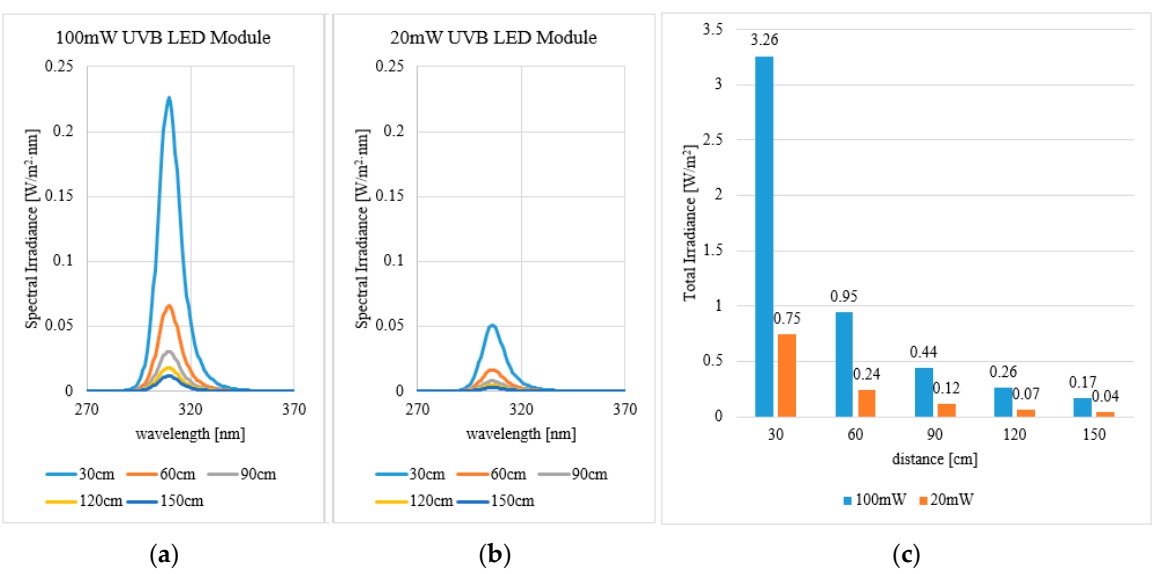

|  (a)  |  (b)  |  (c)  |

**Figure 3.** Irradiance and spectral irradiance by distance step. (**a**) Measured results per wavelength increments: 100 mW UVB LED module; (**b**) measured results per wavelength increments: 20 mW UVB LED module; (**c**) irradiance calculation result by distance step.

Figure 3a,b show the results of the 20 mA UVB LED and the 100 mA UVB LED modules measured. Their irradiance decreased as the distance from the lamp increased. As expected for any light source, the irradiance of the 20 mW UVB LED module was 0.075 W/m$^2$ and 0.004 W/m$^2$ at 30 and 150 cm distances, respectively. The irradiance of the 100 mW UVB LED module was 0.326 W/m$^2$ and 0.017 W/m$^2$ at 30 cm and 150 cm distances, respectively. Further, considering the general lighting environment, the distance between the measurement equipment and the lighting was set to 150 cm; irradiance was then measured according to the adjustment of the applied current. Through increasing the applied current from 50 to 350 mA by 50 mA over seven phases, their UV characteristics were measured and analyzed. The results are shown in Table 4.

**Table 4.** Irradiance of the UVB LED light source modules by applied current. (distance: 150 cm) (unit: W/m$^2$).

| Applied Current (mA) | 20 mW UVB LED | | | | | 100 mW UVB LED | | | | |
|---|---|---|---|---|---|---|---|---|---|---|
| | UV | UVB | UVA | EUVB | EUVA | UV | UVB | UVA | EUVB | EUVA |
| 50 | 0.00064 | 0.00053 | 0.00011 | 0.00015 | 0.00000 | 0.00282 | 0.00218 | 0.00064 | 0.00040 | 0.00001 |
| 100 | 0.00131 | 0.00111 | 0.00020 | 0.00032 | 0.00000 | 0.00564 | 0.00439 | 0.00125 | 0.00080 | 0.00001 |
| 150 | 0.00196 | 0.00167 | 0.00029 | 0.00048 | 0.00000 | 0.00829 | 0.00646 | 0.00183 | 0.00118 | 0.00002 |
| 200 | 0.00262 | 0.00223 | 0.00039 | 0.00064 | 0.00000 | 0.01077 | 0.00837 | 0.00240 | 0.00152 | 0.00003 |
| 250 | 0.00324 | 0.00277 | 0.00047 | 0.00079 | 0.00001 | 0.01308 | 0.01012 | 0.00296 | 0.00182 | 0.00003 |
| 300 | 0.00386 | 0.00330 | 0.00056 | 0.00094 | 0.00001 | 0.01527 | 0.01173 | 0.00354 | 0.00210 | 0.00004 |
| 350 | 0.00449 | 0.00383 | 0.00066 | 0.00108 | 0.00001 | 0.01733 | 0.01322 | 0.00411 | 0.00234 | 0.00005 |

Although both modules used UVB LEDs, some of the optical characteristics of UVA were measured. However, the EUVA was close to 0 when the influence on the human body was weighted. Overall, it is expected that there will not be any harmful effects on the human body by UVA. In addition, the output of the applied current was proportional to the irradiance of UVB and EUVB, and it was confirmed that controlling the applied current can provide the required irradiance. The types and number of UVB light sources needed to support the synthesis of vitamin D were then selected. A previous study suggested that a EUV dose of 105 J/m$^2$ is required to meet 200 IU of vitamin D when 6–10% of the body (of Korean individuals) is exposed [26]. Moreover, the synthesis of vitamin D increases in proportion to the exposed area of the body. Thus, in this paper, the target UVB dose was calculated as 70 J/m$^2$, assuming that the area of the human body that was exposed in the room was 9–15%. When a current of 350 mA was applied to 20 mW and 100 mW UVB LED light sources for 16 h, a EVUB dose of approximately 63 and 138 J/m$^2$ was provided, respectively. Therefore, in order to provide a EUVB dose of 70 J/m$^2$ or more, four 20 mW UVB LEDs were selected as UVB light sources. Figure 4 shows the general UVB LED indoor lighting proposed in this study. Figure 4a shows both sides where four UVB LEDs with a 20 mW output specification are attached and where multi-channel white LEDs for a brightness of 500 lux are placed.

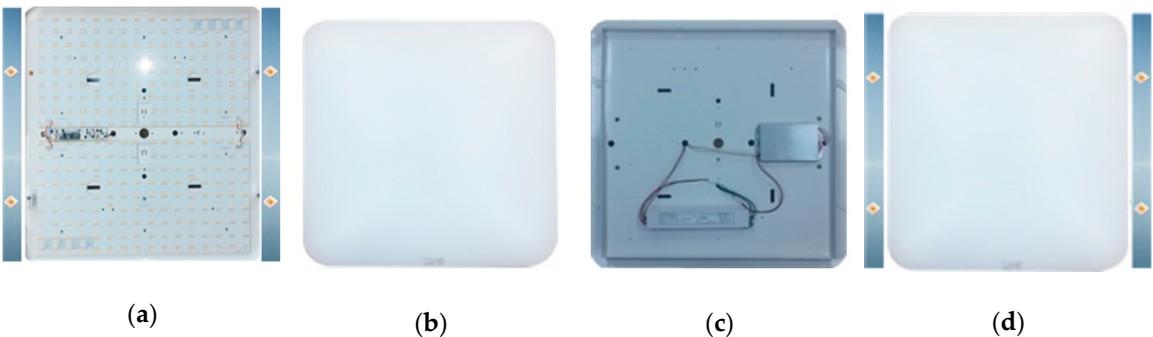

(**a**)      (**b**)      (**c**)      (**d**)

**Figure 4.** General lighting of indoor UVB LEDs. (**a**) vis and UVB LED array; (**b**) diffusion plate; (**c**) control & power; (**d**) front (apply diffusion plate).

Figure 4b is a diffusion plate for a visible LED having a transmittance performance of approximately 66%, and Figure 4c is a controller and a power supply module for controlling brightness. An AC/DC converter with a 47 V output performance was applied to the power supply module, and the controller was implemented to have up to DC 27 V of output through the LED channel. In addition, Figure 4d shows the front profile of the system after implementing a diffuser plate. Since UVB cannot penetrate the diffuser plate, it was made to cover only visible LEDs.

## 4. Results and Discussion

The environment for measuring the optical characteristics of lighting was set as shown in Figure 5. First, a lighting box with a volume of 120 × 120 × 200 cm was used to cut off extraneous light, and a light was set in the upper part of it.

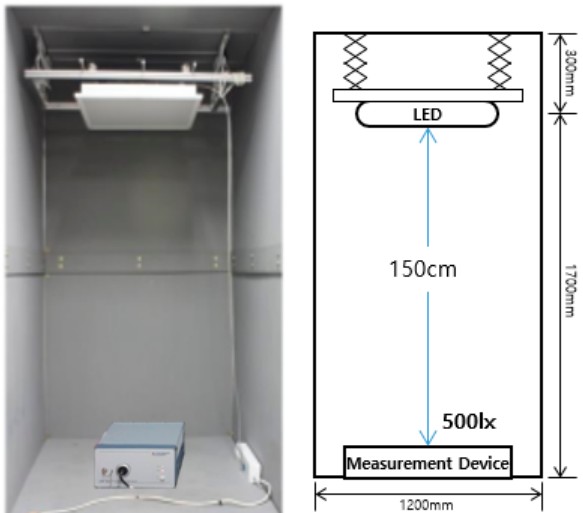

**Figure 5.** Environment for measuring optical characteristics.

In the lower part, a spectral radiometer was installed, and the distance between the light and the spectral radiometer was kept at 150 cm. The brightness of the light was adjusted to 500 lux. The spectral radiometer used was a CAS 140CT device capable of measuring wavelengths in the range of 200–800 nm. Figure 5 shows an environment for measuring the optical characteristics of the proposed lighting.

In this experiment, to confirm the optical characteristics of the proposed lighting, one kind of commercial LED lights was measured and compared together, and the results are shown in Figure 6.

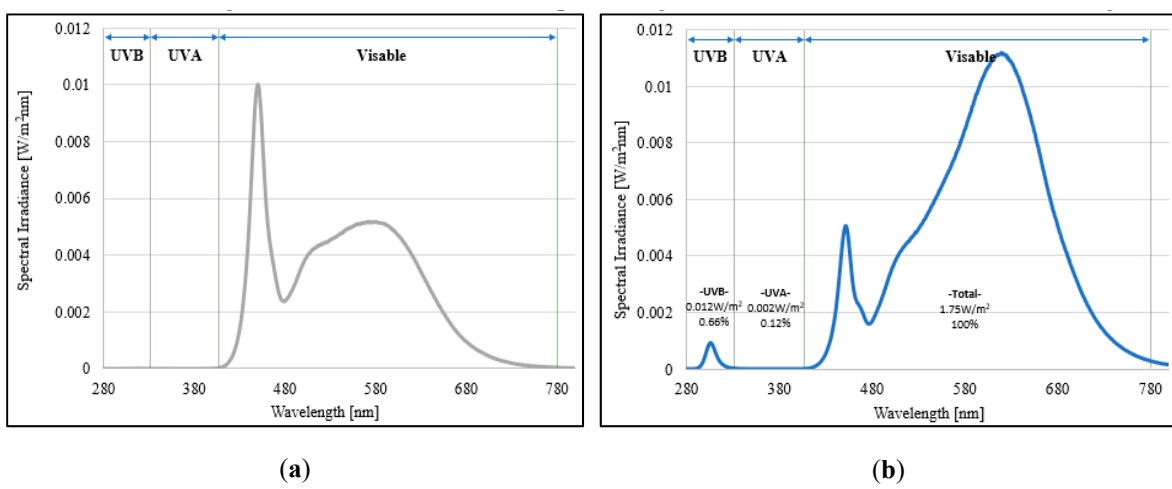

(**a**)            (**b**)

**Figure 6.** Spectral distribution of commercial and proposed general lighting. (**a**) General (normal) lighting; (**b**) proposed lighting (UV and visible).

Figure 6a shows the result of the measurement of LED lighting (merlot lighting, 655 × 318 × 83 mm) on the market that can be mounted on ceilings, showing the spectral irradiance of general lighting. Figure 6b is the proposed general UVB LED indoor lighting. It shows the measurement result when 200 mA is applied after applying four UVB LED light sources with the 20-mW output standard. Unlike the general lighting in Figure 6a, the proposed lighting in Figure 6b had a spectral distribution in the

UV (280–400 nm) band. As a result of the experiment, the total irradiance of the proposed lighting of the UV and visible light had an irradiance of 1.75 W/m². The EUV on which the erythema-weighted function of Equation (6) was applied in the spectral distribution in Figure 7b was calculated to check if erythema occurred on the skin. The calculated EUV was 0.00325794 W/m², which is equivalent to 1/200 of that of natural light as shown in Figure 2. When exposed to the proposed lighting system for 16 h, the EUV dose was about 187.66 J/m², which would not cause erythema for general skin types (Skip Types 1–6) [14]. In addition, a chemical hazard weighting function was applied for the UV zone to check the hazard on the human body by UVB-LED, with the results illustrated in Figure 7.

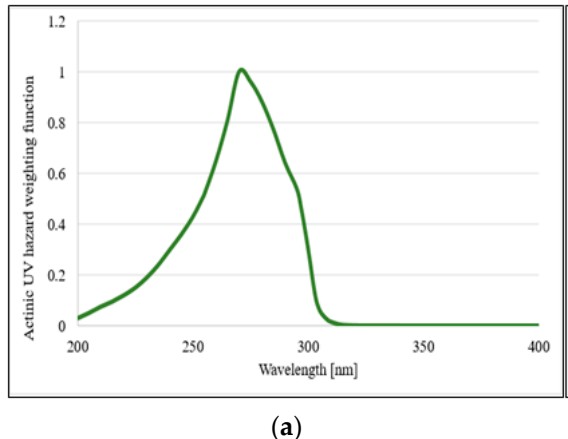　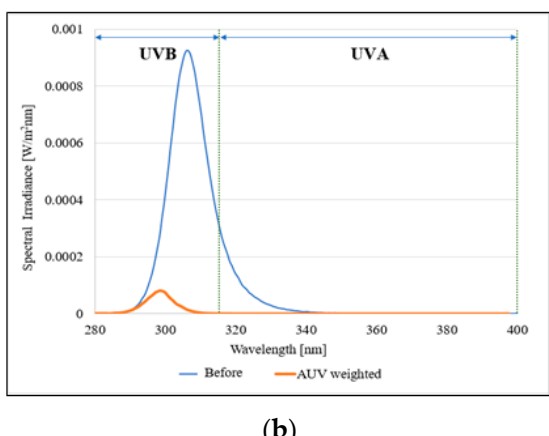

**(a)**　　　　　　　　　　　　　　　　　　**(b)**

**Figure 7.** Spectral distribution of the proposed lighting (UV) and actinic hazard weighting function. (**a**) Weighting function of an actinic hazard; (**b**) Before and after applying the proposed lighting and the actinic hazard weighting function.

Figure 7a shows the actinic hazard weighting function as well as the weighted value approaching the maximum of 1 at 300 nm or less. Figure 7b shows the spectral distributions before and after the application of the actinic hazard weighting function to the UV area of the proposed lighting. When the actinic hazard weighting function was applied, the hazard distribution was adjusted to the range of 280–320 nm from the existing UV wavelength range. The results of applying this to the AUV and NUV formulas (Equations (6) and (7)) are shown in Table 5. The measured values of AUV and NUV of the proposed lighting were 0.001 W/m² and 0.002 W/m², respectively, which are lower than the reference values of 0.001 W/m² and 10 W/m². Their hazard ratings are all considered Exempt, which is the safest rating.

**Table 5.** Comparison of the IEC 62471 standard and UV general lighting measurement results.

| Item | Reference Value | Result Value after Measurement | Hazard Rating |
|---|---|---|---|
| Actinic ultraviolet hazard (AUV) | 0.001 W/m² | 0.001 W/m² | Exempt |
| Near-UV hazard (NUV) | 10 W/m² | 0.002 W/m² | Exempt |

We also applied the vitamin D synthesis amount (Equation (8)) derived from the previous study to confirm whether the synthesis of vitamin D is supported by the EUVB dose of the proposed lighting [33].

$$Vitamin\ D = \frac{\text{EUV}\left[\text{W/m}^2\right]\ \times\ \text{Etimes [s]}\ \times\ \text{Earea [\%]}\ \times\ 40\text{IU}}{\text{MED}\left[\text{J/m}^2\right]} \tag{8}$$

At this time, the expected amount of vitamin D synthesis was calculated by applying 0.003245 W/m² of EUVB irradiance, 16 h (57,600 s) of exposure time (Etimes), Skin Type 3 (Korean), and 15% of the exposed area (face, neck, hands, and arms) for the proposed lighting. As a result, it was confirmed that 374 IU of vitamin D can be generated by providing an EUVB dose of approximately 187 J/m²

through the proposed lighting. The Korean Nutrition Society recommends a daily vitamin D intake of 200 IU per day for individuals aged from 20 to 49 and 400 IU for the remaining age groups [34]. Therefore, it was confirmed that it is possible to support the synthesis of vitamin D in the body through the proposed general indoor lighting.

## 5. Conclusions and Future Research

This study proposes general indoor UVB LED lighting that produces vitamin D for the human body by reproducing solar UV characteristics. UV ray intensity was measured in summers (noon), and the characteristics of solar UVB were confirmed. To provide a safe level of UVB light, we measured and analyzed, based on distances and applied currents, the irradiance of two kinds of UVB LED light sources of 20 and 100 mW output specifications that could control the output of the applied current. To provide the UVB dose via indoor lighting, the application plan of each UVB LED light source was derived. We proposed general indoor UVB LED lighting with four 20 mW UVB LED light sources. The proposed lighting was controlled to have a brightness of 500 lux at a distance of 150 cm. EUVB irradiance of 0.00266 $W/m^2$ was provided by applying a current of 200 mA to the UVB LED light source. The proposed lighting was then measured using a spectral radiometer (CAS 140CT) in a lighting box in which extraneous light was cut off. Additionally, a chemical hazard UV light item AUV and near-UV light NUV, which are the UV rays' hazard items of IEC 62471, were calculated. AUV and NUV were shown to be 0.001 and 0.002 $W/m^2$, respectively; thus, both were found to be in the safest grade, Exempt. Moreover, the EUV calculation value to find out the incidence of erythema by UV was 0.003245 $W/m^2$. If a person were exposed to the proposed lighting for 16 h, the EUV dose would be about 187.66 $J/m^2$, which would not cause erythema for general skin types (Skip Types 1–6). Furthermore, if the vitamin D calculation formula were applied as in the previous studies, a EUVB dose of about 187 $J/m^2$ would be provided to produce 374 IU of vitamin D, provided that an Asian (Skin Type 3) exposing 15% of their body were exposed to the proposed lighting for 16 h. This suggests that general indoor UVB LED lighting that can synthesize vitamin D in humans by providing safe UVB light using a UVB LED light source that can control the applied current should be developed.

In the future, research will be required to build a lighting control system that works with UV smart devices to provide users with an optimal UVB dose via UVB LED general lighting and to offer UVB lighting services for individual users. In addition, it is necessary to validate the safety of the proposed system by conducting a clinical test for more varied skin types and by cooperating with the experts of photobiology and dermatology, and to commercialize the proposed technology to resolve health issues such as vitamin D deficiency.

**Author Contributions:** Investigation, methodology, and writing of the original draft: S.-T.O. Conceptualization and supervision: J.-H.L. Investigation: D.-H.P.

**Acknowledgments:** This work was supported by the National Research Foundation of Korea (NRF) grant funded by the Korea government (MSIP) (No. 2017R1A2B2005601). The research was supported by the International Science and Business Belt Program through the Ministry of Science and ICT (2015-DD-RD-0068-04).

**Conflicts of Interest:** The authors declare no conflict of interest.

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
