# Peer review of "Designing Safe General LED Lighting that Provides the UVB Benefits of Sunlight"

_applsci, doi:10.3390/app9050826_

Round 1
Reviewer 1 Report
Good review work, the manuscripht has improved in a relevant way.
In table 3 the heather and the contecnt of the last column are incoherent (350mA are current) It is a mistake or misunderstand.
Best regards, good work
Author Response
Dear Reviewer1.
I sincerely express my gratitude for your review which helped in improving the quality of my paper.
I have reviewed papers for which you have given suggestions and made the following corrections among the points.
1. In table 3 the heather and the contecnt of the last column are incoherent (350mA are current) It is a mistake or misunderstand.
[After revision - The inaccurate terms and units were corrected.]
- Spectral resolution[nm] -> Maximum Current[mA]
I sincerely appreciate your review of my paper.
Reviewer 2 Report
I have reviewed the modified manuscript, and I am happy to see many of the errors have been corrected, however I still have concerns about the manuscript. Particularly in that the authors have answered queries about reviewers’ critiques within their responses, but not thought to add that information to the manuscript as justification for their work presented here.
For example, in response to Reviewer 1 comments about why vitamin D supplementation is not discussed as a reasonable alternative to this project, it is explained by the authors that sunlight is best and that many of the population in the authors home country might not be able to gain access to solar exposure, or experience hazards due to being outside.
None of this response is overly discussed in the manuscript which is disappointing. The same goes for the comments regarding dust particulates, which appears to be a major factor, but is not discussed much at all. This sort of justification is crucial for demonstrating a need for your proposed project, yet it is barely mentioned. However, the ratio of UVB to UVA varies with solar elevation, and effective erythemal exposure does not always match up to effective pre-vitamin D exposure. I recommend you review the following papers to consider the issues between vitamin D effective dose and erythemal effective dose:
Felton, S. J., Cooke, M. S., Kift, R., Berry, J. L., Webb, A., Lam, P. M. W., ... Rhodes, L. (2016). Concurrent beneficial (vitamin D production) and hazardous (cutaneous DNA damage) impact of repeated low-level summer sunlight exposures. British Journal of Dermatology, 175(6). DOI: 10.1111/bjd.14863.
Webb, A. R., Farrar, M., Kift, R., Durkin, M., Allan, D., Herbert, A., ... Rhodes, L. E. (2013). Targeted sun exposure guidance for South Asians living at northerly latitudes could assist avoidance of vitamin D deficient status. J Investigative Dermatology, 133, S160
Farrar, M. D., Webb, A. R., Kift, R., Durkin, M., Allan, D., Herbert, A., ... Rhodes, L. E. (2013). Efficacy of a dose range of simulated sunlight exposures in raising vitamin D status in South Asian adults: Implications for targeted guidance on sun exposure. American Journal of Clinical Nutrition, 97(6), 1210-1216. DOI: 10.3945/ajcn.112.052639.
Webb, A. R., Kift, R., Berry, J. L., & Rhodes, L. E. (2011). The vitamin D debate: Translating controlled experiments into reality for human sun exposure times. Photochemistry and Photobiology, 87(3), 741-745. DOI: 10.1111/j.1751-1097.2011.00898.x.
Also, this group has also been able to demonstrate that a person of types 2 -4 skin can obtain sufficient vitamin D dose, within a restricted time period over the lunch hour, thus reducing environmental exposures. I strongly recommend you review this research to complete your literature review for this paper.
This follows on from the earlier comment about lack of explanation in the paper. Reducing exposure time at the peak exposure point of the day is a good way to reduce exposure to other particle hazards, although its not fully explained in the manuscript what these hazards constitute or how widespread they are in their country. Statistics from a national body would be extremely useful to be provided in this case.
Additionally the argument that supplements do not provide “enough” vitamin D is not sufficiently argued by just including the articles referenced, as many more articles suggest that supplementation can be achieved, with even particularly high levels without health concerns. Further justification as to why artificial radiation exposure is recommended for the literature review.
A number of the reviewers commented on the necessity of the inclusion of the measurement of the solar irradiance for a single day. It would be more useful for the authors to instead saying that it would be “advantageous” to identify the characteristics of solar UV irradiance, I would have thought it would be more useful to indicate that this is an example of a maximum irradiance possible in summer. The authors note that the measurement was taken on the top of a 10 story building which may suggest that irradiances will be slightly higher than at ground level. The authors should also indicate what the ozone and aerosol measurements were on that particular day to provide a complete array of information about typical solar irradiance in that particular location.
It would be understandable then if that information was used to determine the proposed exposure for the skin type, yet the authors obtain that value elsewhere and do not require the solar exposure for that important information.
In response to reviewer 4, the authors request the reviewers confirm that the solar irradiance seems typical, however that is not the reviewers’ role here. It is up to the authors to show that the solar irradiance is typical, and that the information provided is useful and relevant to the paper itself, otherwise it is superfluous information. The authors should carefully consider whether this information does provide any further information that supports the paper’s aims, especially if it is not used for any calculations that support the conclusions of the paper.
While there are many improvements, there are still a number of issues that need correction, including general English expression as well as missing requested information by the reviewers.
Line 14 – replace “many people with vitamin d deficiency” with “there is increasing concerns about low vitamin D amongst individuals”
Line 19 – delete “powers”
Line 26 – this will be addressed again in the results/conclusions, but no total UV dose from this set up for a single day has been provided (or even erythemal dose which is just as applicable)
Line 35 – replace “light” with “radiation” and replace “causes skin cancer” with “is associated with increased risk of developing skin cancer”.
Line 36 – delete “And”. Capitalise “Any”
Line 38 – delete “etc” - it adds nothing to this line
Line 40 – discuss the reasons why there is lack of exposure to UV radiation (see comments provided earlier to answer this (use the explanations you gave to the reviewers).
Line 44 – I don’t agree that this statement is correct (see earlier comments) however you could say that “the amount ingested is considered by some as insufficient to produce appropriate vitamin D serum levels.” It would be preferable that you add additional evidence for this.
Lines 45-46 why are only young people addressed here? what about other age groups. Additionally, it can take less than 4% of the day to achieve vitamin D levels from solar irradiance – this is not a sufficient argument to state that a majority of time spent indoors is a main reason for low vitamin D. Expand this.
Lines 48-50- this section requires more detail – it provides no information that is useful to the reader. Is it expensive? Accessible? What is the main reason for the special lighting not to be usable to the general public?
This would be the ideal point to discuss solaria treatments and indicate why they are not suitable. This thread can then be addressed in the final discussion and explain how this system differs to solaria.
Line 64 and line 75 – I disagree that the word “safe” can be used. I suggest using a different descriptor.
Line 79 – it is confusing to see a percentage used to express a ratio. Correct the sentence. Suggest using something like proportion rather than ratio.
Line 81 – sentence starting with “in order to confirm…” - delete this sentence. Replace with – “For this project, the local solar UV irradiance was measured and analysed to provide a typical solar irradiance used to calculate UV exposure”, or something similar. However, only do so if retaining this section. I am not convinced it is necessary to keep in the manuscript (as discussed earlier). If retaining, how does the roof top measurement compare to ground based measurements – is there any difference? This information may be useful.
Figure 1 – I don’t see the purpose of including percentages that do not agree with the statement you made in line 79 (about 5% of the UV and visible solar spectrum is UV) – I would suggest either deleting line 79 or deleting the percentages you have included in the figure. It doesn’t make sense to include them since you have already stated these could change according to UV irradiance influencing factors.
Line 95 – delete “To confirm”. Capitalise “The…” you are presenting information here, not confirming it.
Line 96 – this definition for the erythema action spectrum is poor. A better definition is “which is the ability of UV radiation to produce erythema at a given wavelength”. Also you should cite the CIE for this definition.
Lines 97-98 – you have not explained why it is necessary to use erythema to estimate vitamin D exposure. Please explain and cite appropriate references.
Line 109 – delete “should be provided based on Asian standard” and replace with “could be provided to skin types commonly found in the Asian region.” – recommend citing the range of skin types of these areas. The first statement suggests that there is only one type of skin type, but surely there is some variation in the skin types amongst the population?
Line 124 – delete “at a distance”
Line 125 – what is the relevance of including the measurement at 200mm with special lighting devices here? This distance is not used anywhere in this project – so either you are missing measurements at this distance, or this information is not necessary and probably doesn’t need to be included.
Line 126 – lux is a measure of illuminance, therefore it is still confusing as to how this is used for general lighting with UV irradiance. It sounds like the general lighting guidelines is unable to cope with including UV irradiance, and therefore different standards may be required. This is worthy of discussion after the results section.
Lines 133-144 – This whole section reads poorly, with written explanations jumping around without stepwise discussion of each equation – instead it mentions equation 4 and equation 7, before equation 5, then fails to mention equation 6. Please fix this section. Also suggest the following: delete “Equation (4) is formula for calculating AUV and NUV, and”. Keep the rest of the sentence and add “(Equation 4)” after “(SUV)” - also make sure the UV is subscripted in the text, as the term “SUV” can mean other things and confuse this section.
Line 146-47 – 500 lux is not a distance - delete “at a distance” – suggest “at an illuminance of” instead. Reviewer 3 queried this issue and the authors stated this was fixed, but it appears to still be an issue.
Line 157 – in the sentence, state that the models are listed in the table. It is also usual to provide where the lamps were obtained from (distributor name and location in brackets).
Table 3 – the last column incorrectly states “spectral resolution (nm)” when it clearly should be current in mA. Please correct.
Figure 3 – the caption for this figure is poorly written and needs more information given it has three individual figures within it. Additionally, if it has three figures then each figure should have its own sub-figure notation, a, b and c. Explain each figure clearly. – suggest to use “per wavelength increment” instead of “distance step”. Figure 3(b) (should be 3c) should have “total” in the y axis to differentiate from the other figures.
Line 172 – Start sentence with “Figure 3(a)” (or “Figure 3 (a & b)”) not “(a)”
Line 173 – delete “as they get further from each other” – replace with “as the distance increases from the lamp, as expected for any light source”.
Line 174 – Insert The” before “Irradiance”
Line 181 – Table caption – include “distance measured at” in the caption. Captions should contain all relevant information and readers should not have to search the text for that information.
Line 184 – the authors state the EUVA was measured also but fail to state why. Please explain. While UVA is not generally known for erythemal damage (except for the 320-340nm wave range) UVA is known for other detrimental issues, such as potential involvement in the development of melanoma, and involved in immune suppression and photoaging. Is the unweighted UVA within a recommended exposure range? UVA output from the lamps is lower than UVB, but that is not even discussed in this paper, which it clearly should be, at least pointing out those factors about UVA. What UVA does to humans is still not well understood and that should also be pointed out. Please add some discussion about this.
Line 187 – replace “kinds” with “types”
Line 201 – again, there was no clarification about the diffusion plate – does it transmit UV equally compared to visible radiation or is there a weighting of the transmission? This was queried in the previous reviews but not answered.
Lines 211-212 delete “At this time,” replace with “the spectral radiometer used was” - delete “was used as a spectral radiometer” at the end of the sentence.
Lines 219-220 – figure 6 caption – (b) change to state “Proposed Lighting (UV and visible)”
Line 224 – Start sentence with “Figure 6 (b)” not “(b)”
Lines 227- 228 – replace “providing UVB light was 1.75W/m2” with “UV and visible light with an irradiance of 1.75W/m2”
Lines 229-233 – this whole sentence is confusing – where does the measurements of “0.039-0.044w/m2” come from and why is the distance different again? If you are comparing against another study, then you should outline what the study was and explain what they were doing.
Also, lines 232-233 is confusing because you state you weight the spectrum erythemally, then apply the actinic weighting on top. This is especially confusing – by applying both, you are not getting the correct hazard to the body at all. This strongly underestimates the exposures actually being received by a person. This should be immediately clarified as it could be presenting incorrect information here.
I wonder if this is showing why you get such low values for actinic exposure. Please confirm how you are obtaining the weighted data, because confounded data could suggest you are not below the exempt levels.
Line 252 – skin type 2.5? – please revise to the most appropriate skin type using the skin typing system currently used as acceptable. State as a range from Type 2 to type 3, not type 2.5.
Line 345 – replace all caps with appropriate upper and lower case text as given by the journal guidelines
Line 349- capitalise “joule” – “Joule”
Also no total UV dose from this set up for a single day has been provided (or even erythemal dose which is just as applicable). A total erythemal dose (not actinic dose) for a 16 hour day from this set up should be provided in J/m2 and possibly even presented in SED (standard erythemal dose) to indicate total hazard to a person. It is essential to know what this value is to understand comparative safety levels -for example, is it sub-erythemal?
Author Response
Dear Reviewer2.
I sincerely express my gratitude for your review which helped in improving the quality of my paper.
The file has been attached to the modification.

Round 2
Reviewer 2 Report
The manuscript seems improved now. The main correction I suggest is presenting the erythemal dose received in the abstract, rather than the irradiance. In fact i wonder if that is a typographical error, since in the body of the text, you state 187W/m^2, but that would be a very high irradiance when that is clearly not the irradiance output you actually measured in the table.
I recommend changing the included sentence in the abstract to indicate exposure and J/m2, not irradiance and W/m2. The authors will also want to revise the section in the discussion on this part also, because it looks like J/m^2 and W/m^2 have been used interchangeably (which is incorrect). The authors should confirm their calculations regarding daily dose before submitting that revision to the editor.
With the provided daily dose, I would like to point out that even though the dose is equivalent to a sub-erythemal for that particular skin type, it is not that far below a minimum erythemal dose for a person with Type 1 skin. Therefore, while general lighting stipulations may make the calculations say that the proposed lighting is in the exempt category, it could still detrimentally affect people with different skin types. Additionally, there are studies to show that chronic cumulative sub-erythemal doses can be detrimental to human health, even of the proposed skin type in this manuscript, regardless of general lighting classifications. I strongly recommend the authors take this into account in future studies and to not just aim for "exempt" categories in general lighting. I also recommend consulting photobiology and dermatology experts before further developing these sorts of devices.
There are some typographical errors in the revised sections, but I am sure the editor can assist with pointing these out.
Author Response
I have attached the file.

This manuscript is a resubmission of an earlier submission. The following is a list of the peer review reports and author responses from that submission.
Round 1
Reviewer 1 Report
The article is focused on using LEDs that produce ultraviolet B radiation, used in combination with other LEDs within the visible spectrum as interior lighting, with the goal to produce non-hazardous exposures to UVB for office workers who do not spend time outdoors therefore not inducing sufficient vitamin D for a health lifestyle.
The paper has explored the merits of this work, however there are number of concerns that the research has not addressed and should be if this is to be a published paper.
I also need to be frank, and advise that I don’t believe using artificial UV radiation exposure is the key to improving human wellbeing by inducing vitamin D. It is also known that sunlight provides other beneficial factors to humans, and that time as short as five minutes of exposure to sunlight every day, can provide positive outcomes to humans. If the paper is to be considered for publications, then the following issues should be addressed, to create a more rounded approach to the problem being addressed, especially for researchers like myself who are averse to the general idea.
Importantly, the paper needs to provide more appropriate terms that are used to quantify electromagnetic radiation exposure.
For example, the term “light quantity” is used repeatedly throughout the paper, and does not make distinction between how the radiation is measured. This has also resulted in some incorrect use of units.
Irradiance is the correct term for power per unit area received from the sun (and the authors use it in some of the figures). The authors should use that term, or radiant flux per unit area as that is more specific also, consistently throughout the paper. The authors might like to refer to it as artificial UV irradiance or similar, but this is a more accurate descriptor than “light quantity” which is generic and non-descriptive. Additionally, when the irradiance is used to calculate UV radiation exposure per unit of time, then the measurement is considering total dose (energy per unit area). Again, UV exposure, or UV dose should be used, not “light quantity” for the same reasons as stated above. Also, because of the usage of such a generic term, errors have been made for example:
Line 96 reports a UV exposure (or UV dose), however it is clearly using the wrong units for dose, which shows irradiance instead of exposure. This is evident by the stated calculation: 0.15W/m2 over 720 seconds provides a solution of 108 J/m2 (not 108 W/m2 as stated). This error should be corrected. More details about corrections about light measurements will be provided later.
The authors use the argument that (for example) many people spend a lot of time outside and do not receive adequate solar irradiance to induce vitamin D production. The authors propose using extremely low UVB irradiance in adjustable artificial lighting systems, to increase vitamin D induction.
A small body of literature has been referenced suggesting that the support for this proposal could be effective by recommending extremely low doses of artificial UVB exposure. However there are three issues that should be addressed by the authors, by consulting further literature and making some comment (either in the literature or the discussion about these factors).
The first is that there is concern that continuous low doses of UV radiation can contribute to specific types of skin cancer development. A suitable starting point to reference to review and discuss is:
Armstrong, B.K. and Kricker, A., 2001. The epidemiology of UV induced skin cancer. Journal of photochemistry and photobiology B: Biology, 63(1-3), pp.8-18.
Some comment should be made in the discussion about whether promoting the use of artificial UV exposure could also be an issue in the causes of skin cancer.
Secondly, there is a potential for the human eye to be damaged by UVB, although the links are not as well established for chronic low exposure, in contrast to acute exposure. Literature surrounding this should be reviewed and included in this paper, indicating the possible hazards that continuous low dose of UVB radiation might have on the human eye. Some examples include but are not limited to:
Sliney, D.H., 2001. Photoprotection of the eye–UV radiation and sunglasses. Journal of Photochemistry and Photobiology B: Biology, 64(2-3), pp.166-175.
McCarty, C.A. and Taylor, H.R., 2002. A review of the epidemiologic evidence linking ultraviolet radiation and cataracts. In Progress in lens and cataract research (Vol. 35, pp. 21-31). Karger Publishers.
Thirdly, many countries have sought to reduce, or completely ban use of solaria (skin tanning lamps). While I understand that this is not the proposed goal of the study, it would be expected in research surrounding artificial UVB lighting, that there would be some discussion in showing how this work is not the same as used in still existing solaria systems, and potentially address how the device will not be similarly used.
I also recommend that the authors review further work by Michael Holick, as many of his studies suggest that just a few minutes a day can be sufficient to induce vitamin D for health purposes.
Line 65 – please define actinic and near UV hazard radiation with wavelength ranges in the text. In fact it would be preferable to see a plot of both weighted functions earlier on in the paper. Only the actinic function seems to be presented later in the paper.
In the methods (section 2 starting line 74) it is stated that the erythemal action spectrum is used. Is it being used as the actinic action spectrum, since an actinic action spectrum is usually more generic in nature (any possible effect)? If this is the case (using the erythemal action spectrum as the actinic action spectrum) then this should be clearly stated in the in the paper. Otherwise, it is confusing as to why the erythemal action spectrum is used, when previous mention of weighted irradiance clearly says actinic which is less specific. If the erythemal action spectrum is being used to represent the actinic action spectrum, then this should be explicitly stated.
Line 77 – I don’t understand the use of the sentence starting “To confirm this…” This statement sounds like you are confirming the effects of sunlight on the human body, when what you are discussing sounds more like you measured the solar spectrum as an example of the study’s location. This whole paragraph will probably need revision, possibly with assistance by a language editor, to ensure correct meaning here.
Figure 3 – The caption information does not match the y axes presented. Please provide the “output performance” described as spectral irradiance in the caption.
Also, the peak irradiance for each scan is provided, in the figure for 750px, however its not clear to me why the peak wavelength irradiance is highlighted when what you are exploring (or what I assume you are exploring) is total irradiance across the spectrum.
Lines 149-152 – the total spectral irradiance values for each distance the light is positioned at is supplied, but not described in this way. This is confusing for the reader, especially if at first they assume that the numbers reported are the peak irradiance, rather than total spectral irradiance measured (or integrated spectral irradiance depending on the author’s preference).
Lines 153-157 – same as above comment, only this should now be referred to as erythemally weighted total spectral irradiance, not “output”
Starting line 151 and onwards throughout the paper – please correct the units of centimetres from “Cm” to “cm”.
Lines 154-155 – the difference between the 20mW and 100mW total erythemally weighted spectral irradiance is curiously at a factor of 2. It would be good to see this discussed as to why this has been observed (how the spectrum doesn’t change despite change in power – is a possible observation).
Table 2 – Column 1 should be more correctly labelled as “Current (mA)” as “hazard items” is generic and non-specific – also it is missing units. Again caption should be more explicit in the description - “light quantity” is a poor descriptor.
Table 2 – additionally the authors should explain their reasoning as to why they explored the erythemally weighted total spectral irradiance for the UVA spectral range only. To what purpose does this measurement provide?
Line 165 – again – change “W/m2” to “J/m2” since this is a dose measurement not irradiance.
Line 166 – the reasoning for why the number of UVB-LED is selected as four is not provided. This may seem obvious to the authors, but your calculation is not obvious to me despite reviewing the results. Please elaborate. You also do not state which wattage is more appropriate, but instead mention in a later sentence. However, the chosen wattage seems to change in different parts of the paper (line 190 says of 20mW, but line 168 says of 100mW type). Which is it?
Line 167 – please confirm that the correct units have been used. I am not convinced these should be irradiance measurements. They look more like dose measurements, but it is hard to tell.
Line 169 – at what distance does this statement apply. Please elaborate.
Figure 4 – please provide some information about the spectral transmission characteristics of the diffusion plate (how much UVB is transmitted?). The statement in the text says that the diffusion plate is for the visible LED. So then is this diffusion plate placed over the UVB LED? What is the general order of layering of these features within the LED device?
Line 181 – please write the full unit of “lux” rather than “_lx”. Also please provide how this value translates to the total erythemally spectrally weighted irradiance. If the lux measurement is visible spectrum only, then this should be pointed out to the reader.
Line 181 – provide company and location of the model number of the spectral radiometer device.
Figure 5 – correct “Cm” to “cm” in figure
Line 186 – why was only one UVB LED used? Is it because only one is available? If not, explain reasoning for only using one type.
Line 188 – what is “M” company? Is this a typographical error, or the actual name of the company?
Figure 6 – make sure the y axis for each part of this figure is consistent. Either use scientific notation or standard notation, but not both. Neither y-axis has a title, only units.
Lines 194-198 – please confirm if you are discussing UV exposure (dose) of UV irradiance in this section. I am not convinced these are all discussing the same type of measurement.
Line 200 – “confirm” is not the correct term to be used here. You measured the spectral output of the UVB/visible LED array and you applied the erythemal action spectrum, and provided the output. That is not confirmation of the information, just presenting the information.
Line 201 – “arear” – I can’t work out what this term is meant to be
Line 204 – you discuss the limits of 0.001W/m2 for the exempted lighting, however its not clear if this calculated value is from total erythemally weighted spectral irradiance or peak irradiance measures from Figure 7.
Line 217 – you present the total vitamin D possibly produced for your scenario. However, to make this value more meaningful, which will also take into account any possible hazards, is to present the total UV exposure (dose) that would be incurred, to achieve that vitamin D dose. Compare the total UV dose to the recommend exposure limits either by WHO or the ICNIRP, or similar standards, to confirm low hazards due the UV exposure. – this relates to Line 232 – because I do not believe you can claim no hazard to UVB exposure, as you have not demonstrated this, due to the missing information requested here.
In fact I recommend you delete the sentence in Line 232 – as I do not believe you can claim this at all. Further discussion on hazards of UVB radiation should be included in the discussion, and summarised in the conclusion. It would not be appropriate to be promoting exposure to artificial UVB radiation and claiming there is no hazard.
If the authors wish to carry out the revisions to the paper for publication taking into consideration the type of readers of the paper, I strongly recommend that they seek the services of a photobiologist or a dermatologist (or ideally both) to confirm they understand how these people would review this paper, and help them to address the sort of concerns I have raised here.
Reviewer 2 Report
This manuscript “General lighting design plan of LEDs that provide 2 UVB characteristics of sunlight” is a study that proposes a comprehensive indoor lighting that can provide to the users a safe UVB light quantity by applying a UVB-LED light source.
I see this manuscript a relevant work, a window to new light applications but with important writing problems, mainly in the use of English and in the order of explanations. I also find difficulties in understanding work because in some cases the explanations are confusing. I note that the work has not been sufficiently revised, the authors have not put together to make a detailed review of all the work and that is seen in important problems of coherence with the treatment of the magnitudes, which are well treated in one sections and bad in others.
If the authors do not develop a complete review of all the writing in the next version, it is impossible an acceptance of the manuscript.
There is weakness in the results explanation, because the units and comprehension of magnitudes are wrong.
This are the problems that I see in the manuscript.
Abstract. “ most people have vitamin D deficiency” This is not the conclusion of your introduction, it is a problem for the people without exposition to sun light but “most” it not true.
“that can provide users with safe UVB light quantity by applying a UVB-LED light source that controls the output of light quantity. “. Who are the “users”?. This is address for office job or for treatment, concrete. Furthermore, the phrase it is very confuse specially “by applying a UVB-LED light source that controls the output of light quantity”, there are not a light control, the authors experiment and define the current for a LED type for obtain the light adequate for the “user”. Finally, the term “light quantity” it is used frequently during text and refer to anything “measuring luminous flux per unit area in lux=lm/m2” or Irradiance of UVB W/m2 or Radiant Energy W·h/m2 . This produce a lot of ambiguity in the text. Any scientific text will be coherent and precise with the use of words.
“The light source combination of UVB-LED that complies with the UV hazard standard of IEC-62471 is derived”. The light source combination of white LEDs and UVB-LEDs, that is to say, the proposed lighting module …..
Line 68: “The spectral irradiance of general lighting based on UVB-LED light sources that produce 500lx, which is a luminance standard of general lighting specified in IEC 62471,” the phase tell literally that the authors propose a 500lx based on UVB-LED, only UVB-LED and some exceptional LED, that can be best explained. The authors propose a 500lx white lamp that additionally have UVB-LED for vitamin D purpose, this implies complying with the standard …..
Line 94. “Therefore, to produce the UV characteristics of sunlight, a UVB accumulated light quantity of ≥108 W/ m2 is required.” The W/m” it is not “accumulated” light, because accumulated light it is energy, that is the instant radiation level at moon. What it is the origin of 108?. This can be 108 W·minute/m2 for day, week or month or this is other subject?, can the authors explain or correct this expression?
Line 92. A previous study revealed that Asians (Koreans) could produce 200 IU of vitamin D when exposed to an EUVB intensity of 0.15 W/m2 for ~12 min (720 s) [21]. Both terms (look previous comment) are correct intensity light of 0.15 W/m2 during 12 minutes, that is intensity power product by time, this is accumulated light energy, E=P*t, but it is need know if this energy it is the result for a day, week or month necessities for obtain an understandable conclusion.
Line 142 to 145 paragraph it is incoherent with the information and figure 3. In Figure 3 there are “Output performance of UVB-LED light source. “ I think this is correct. But in the paragraph “ For the experiment, two test lighting modules combining 10 light sources are 145 developed and arranged at intervals of 30 cm each from a distance of 30~3750px; the optical 146 characteristics at each step are measured, and the results are shown in Figure 3.” This phrase refer to “two test lighting modules combining 10 light sources “ as Figure 3.
There is a problem in the text information because finally I do not understand if Figure 3 it is only the spectral performance of one UVB-LED or the performance of the four UVB-LED of the lamp, and the Figure do not indicate the current applied for the LED or LEDs. The authors can first explain the UVB-LED characteristic and later the lamp not a mix.
Line 150. What it is the current applied and the number of LEDs for Figure 3 and this paragraph “The 20mW UVB-LED light source module shows 0.74 and 0.04 W/m2 output performance at distances of 30 and 150Cm, respectively [The reviewer does not see this data in Figure 3, probably are the integration of the complete LED spectre or other thing, please explain]. The 100mW UVB-LED light source module shows 3.24 and 0.17 W/m2 output performance at 30 and 150Cm, respectively [ the same problem] ”. And finally Cm it is not correct.
Line 159 and next Table 2. Light quantity will be light power measured at 3750px it is of 1 UVB-LED or four UVB-LED in the lamp position by applied current, the current it is only for the four LEDs with the white LED OFF or for all LEDs?. The term Hazard Items on the table, is it not the column of current applied?.
Line 103 “Each item measures the radiance and the radiance intensity based on the wavelength range; then, eight hazard items are rated as exclusion, low-risk, moderate-risk, and high-risk.· If there are eight hazard items because the authors only enumerate four.
Paragraph 164 “In the case of Koreans presented in previous studies, they require a UVB light quantity of 108 W/m2 to meet 200 IU of vitamin D [17] and therefore, the number of UVB-LED light sources is set to four “ There are several errors and inaccuracies in this paragraph.
· 108W/m2 it is not a light quantity it is an instant light power. Can be 108W”min/m2.
· “It is confirmed that UVB light quantity up to 220 and 761 W/m2 can be provided when four 20 mW UVB-LED light sources and four 100 mW UVB-LED light sources are applied and the power has been applied for 16 hours”. If the authors provide a electrical power that produce (20 and 100) that produce a light power (table 2, I suppose) during 16h this is an accumulated energy during this time of operation no other power of 220 and 761 a W/m2 respectively, E=P*t, This is very basic error.
Result section. The same problem with light quantity, identification of power per unit of surface and energy per unit of surface and day, week, moth, …. and furthermore the same problem in the Conclusions.
Paragraph in line 194, the values are of power of accumulated UVB light energy.
Paragraph in line 214, The authors explain that one day of sun light can be replace for 16h of UVB-LED light, it is that?. But for what UV-LED, what current. This is the last paragraph of the result and it is not concrete with the condition of the results.
The results section it is difficult of understand because has different low related sentences in the same paragraph, the authors will precise and explain the results, probably with work bullet will be more comprehensible.
Line222. “UVB-LED light sources of 20 and 100 mW output specifications that could control the output of the applied current”. I read the light sources could control the output of the applied current, I this that the authors want to means that the light power of the LEDs can be controlled by current.
Line 225 “We proposed UVB-LED indoor general lighting with four 20 mW UVB-LED light sources.” Why do you proposed this UVB-LED and not the 100 mW LED, where is the explanation?, I am loss this explanation in the text.
Concrete the goal of this lamp, that type of users, how much hours of exposition are need for obtain the D vitamin objective, with that UVB-LED, the distance and LED power.
For help with the clarification of my explanations the authors can see this paper with other goals and specific area, but that use UVB-LEDs. Kalajian, T. A., Aldoukhi, A., Veronikis, A. J., Persons, K., & Holick, M. F. (2017). Ultraviolet B Light Emitting Diodes (LEDs) Are More Efficient and Effective in Producing Vitamin D 3 in Human Skin Compared to Natural Sunlight. Scientific reports, 7(1), 11489. They use a unit of energy for identify the necessities of human skin for previtamin D3, from the text “To determine which LED was most efficient in converting to previtamin D3, the energy output for each LED was measured using the Solarmeter to calculate the exposure time needed for each LED to reach 46.8 mJ/cm2”. In other paragraph “It took only 0.52 minutes for the 293 nm LED to emit 11.7 mJ/cm2 which resulted 1.2% of the 7-DHC to be converted to vitamin D3. Exposure to the same amount of energy from the sun took 32.5 minutes and only 0.5% of the 7-DHC was converted to vitamin D3. “ in this case they use as energy unit per unit area mJ/cm2, but the authors by coherence will be use Wh/m2 of Wmin/m2. I do not have any personal interest in this paper.
Reviewer 3 Report
Reviewer.
In General
This article presents a topic of great scientific interest and technological innovation as a scientific article could be perfectly published in this journal from the point of view of this reviewer, once a more in-depth review of it is carried out. There are aspects that could clearly be improved to make the article more scientific and impactful. A major revision is proposed.
The introductory part is well formulated but perhaps it would be advisable to reinforce the bibliographic part with articles of greater relevance and proximity in time. It is advised that all acronyms be defined beforehand or at least the first time they appear in the text.
The methodology must be exposed much more clearly to allow a better follow-up to the reader. The method and equipment used must be better explained, as well as the selected light sources. It is recommended to use Flowcharts or other graphic material that allows a better and better understanding. It is also recommended to use more Tables and Figures that allow comparing the characteristics of the two LED lamps with which the study used in the experiment is concluded. Defining more precisely its particularities. It is recommended that the measurement equipment used in the experiment be technically described in tabular form. Indicating accuracy, range, tolerance, etc ...
A complete restructuring of the research process and method is recommended, explaining more clearly and precisely the applied research process. Supporting similar studies and, if possible, more up-to-date, defining precisely the state of the art, explaining better the Method and Tools used that justify the tested models and give a more real and rigorous feeling to the experiments carried out.
The bibliography is scarce and the references must be more precise and adjusted
In particular
Next we will describe some aspects that this Reviewer understands that could be improved:
Paragraph 50 to 53- You should justify this statement with bibliographical references to related studies.
Paragraph 75 to 80. You must explain the Method better and in more detail as well as justify the sampling point. In addition you must add Measurement Equipment used for the experiment and the technical specifications of the measurement equipment used summarized in a Table
Line 85. You must indicate the Report of the CIE that justifies this affirmation
Paragraph 90 to 95. You should justify this statement with a mathematical calculation that explains and clarifies it.
Line 99 to 100. This quote [13] does not correspond in the Bibliography. It should be revised.
Paragraph 108 to 114. It is recommended to refer to a standard or International Standard that allows the Assessment and Management of Risks in Projects in general Standard UNE - ISO 31000: 2010. Risk management. Principles and guidelines. Edition (2010) Committee CTN 307-Risk management.
Line 118. All acronyms must be defined at least the first time they appear in the text. In addition, they must justify their application in this case study.
Line 125. This study is overused for this appointment. The Method should be strengthened with other references to similar applied studies.
Paragraph 130 to 138. In this expression it speaks of distance ... but it measures illumination in lux ... it should clarify this affirmation. Here is another incongruity. Describe Brightness and bring Iluminacia en lux Should clarify and explain this paragraph better to not confuse the reader. Could you clarify the type of driver used? Characteristics of it, and operation.
Figure 2. This figure provides little and is very simplistic. It would be more interesting a detail plan that describes the place or a table with the characteristics of the place. The photograph on the right does not look good and neither does it know what information it brings. It should be clarified better what is intended to indicate or represent. On the other hand it seems that 500 lux is an excessive illuminance for a safe home or work environment. It must be justified and clarified.
Paragraph 142 to 147. The reason why these light sources have been selected and not others should be better justified. References to studies that have done so would be enough. It would be very interesting to evaluate other parameters such as the Luminace cd / m2 or the RGB of each of the lamps used in the Study. (x, y) in the color abacus or even the color temperature of the lamps in Kelvin ºK.
Line 149. This is really something very obvious. And what could you expect? It should be deepened a little more
Table 2. The current does not appear in this Table. It should be included.As well as Illuminance in lux and luminance in cd / m2.
Figure 4. It would be very interesting an electrical diagram of the solution defined in the photographs, providing in a table the technical characteristics of all the equipment of the photo.
Line 179. It is very important to indicate in a Table the features of the Spectrophotometer or spectral radiometer used in this experiment. Class, accuracy, range of work, etc ... ..
Paragraph 186 to 191. It would be very interesting to define in detail the characteristics of the commercial source used, to strengthen the results of the experiment and make them more real.
Line 204. These figures (Figure 7) should be better explained. They are really very interesting but for example we do not see the values of the NUV represented.
Paragraph 214 to 217. Could this experiment in Asia really be applied in the rest of the world? Could you justify it?
The Conclusions, although very interesting, are accurate, but should be better explained and synthesized.
The Bibliography should be revised with references to Studies or publications more appropriate and updated in time to completely improve the article and reinforce the Methodology and Final Conclusions.
We appreciate the efforts of the authors of the study and consider this work very interesting, although we propose a thorough revision of it. Major revision.

Reviewer 4 Report
The manuscript titled "General lighting design plan of LEDs that provide UVB characteristics of sunlight" presents the characterization and the dimensioning of a luminaire using common white LED along with special UVB LED chips. The idea is relatively novel, and the approach of the research is presented in a good way. The manuscript can be revised in the following points in order to be more complete and to present the results of the study in a clearer way.
1. The research is based on the IEC 62471 which is related to photobiological hazard of lamps and lamps system. However, the illumination system that the authors proposed must be justified why it is considered as a GLS (General lighting system) or not according to the IEC standard.
2. In several points inside the document is written about the distance of 500lx. Lux is illuminance level and not distance. As per IEC standard test distance should be the corresponding distance where average illumination is 500lx. Therefore, text must be updated accordingly.
3. Section 2 is based in one-time measurement of the sky spectrum and the corresponding calculation of UVB quantity is strongly depends on this spectrum. Authors must justify why this single measurement can be act as the reference spectrum. The should also add different sky spectrums (real measurements or reference spectrums) in order to justify the 108W/m2.
4. Line 68. 500lx is not luminance but illuminance. Please consider the same correction throughout the text
5. Figure 3 could be replaced with a line plot (LED chips distance Vs Total radiance output) in order to be more informative about the effect of LED separation. An additional single LED spectrum at a random distance is although needed for the presentation of the spectrum of the UVB LED.
6. Authors are using the phrase optical characteristics in a misleading way in many points in the text. The characteristic that they are measuring is the radiance/irradiance output. Optical refers to optics and light distribution. Please consider the update of these phrases.
7. Light is visible radiation. Anything below ~400nm and over ~800nm can not be called light. Therefore, when authors refer to UVB they have to use the word radiation.
8. Table 2 must include the term “LED driving current” or something similar under the term “Hazard items”. Otherwise is not so clear what the first column stands for.
Good luck.